# Microstructure of Plasma Nitrided AISI420 Martensitic Stainless Steel at 673 K

**Tatsuhiko Aizawa [1],\*** , **Tomoaki Yoshino [2]**, **Kazuo Morikawa [3]** and **Sho-Ichiro Yoshihara [4]**

1 Surface Engineering Design Laboratory, Shibaura Institute of Technology, Tokyo 144-0045, Japan
2 Komatsu Seiki Kosakusho, Co., Ltd., 392-0012 Suwa, Japan; yoshino@komatsuseiki.co.jp
3 Tokyo Metropolitan Industry Research Institute, Tokyo 144-0045, Japan; morikawa.kazuo@iri-tokyo.jp
4 Department of Engineering and Design, Shibaura Institute of Technology, Tokyo 108-8548, Japan; yoshi@sic.shibaura-it.ac.jp
\* Correspondence: taizawa@sic.shibaura-it.ac.jp; Tel.: +81-3-6424-8615

**Abstract:** Martensitic stainless steel type AISI420 was plasma nitrided at 673 K for 3.6 ks to investigate the initial stage of the nitrogen supersaturation process without the formation of iron and chromium nitrides. SEM-EDX, electron back-scattering diffraction (EBSD), and TEM analyses were utilized to characterize the microstructure of the nitrided layer across the nitriding front end. The original coarse-grained, fully martensitic microstructure turned to be $\alpha'$- $\gamma$ two phase and fine-grained by high nitrogen concentration. Below this homogeneously nitrided layer, $\alpha'$-grains were modified in geometry to be aligned along the plastic slip lines together with the $\alpha'$ to $\gamma$-phase transformation at these highly strained zones. Most of these $\alpha'$-grains in the two-phase microstructure had a nano-laminated structure with the width of 50 nm.

**Keywords:** low temperature plasma nitriding; martensitic stainless steels; nitrogen supersaturation; fine two-phase structuring; nanolamination in $\alpha'$-grains

## 1. Introduction

A martensitic stainless steel type AISI420 and its derivatives have been widely utilized as a mold material for injection molding. Various heat treatments and surface modifications were applied to improve the wear resistance and corrosion toughness of these stainless steels, as well as the nitrogen bearing iron alloys [1]. High nitrogen concentration provides a promising approach to accommodating the high strength and toughness of the stainless steels [2]. No nitride precipitation in these nitrogen steels becomes a key point in materials design in order to simultaneously increase the fatigue strength and toughness [3]. In addition, two-phase structuring is also preferable for improving the material performance in practice [4]. However, as pointed out in [5], a higher nitrogen solute concentration than 1 mass% is too difficult to improve the properties and the performance of higher nitrogen stainless steels in practice.

A plasma nitriding process below 700 K makes nitrogen supersaturation into stainless steels with a higher nitrogen solute content than 1 mass% and without nitride precipitates [6]. This low temperature plasma nitriding was first characterized by the formation of the nitrided layer with a thickness more than 80 μm—even at 673 K for 14.4 ks [7,8]. Large $\alpha'$-phase peak shifts, as well as new peaks for $\gamma$-phase, were detected by XRD analysis to prove that $\alpha'$-lattices expanded in the c-axis and elastically strained enough to make the transformation to $\gamma$-phase [9–11]. CrN and $\gamma'$-Fe$_4$N were not detected in the trace level; the detected $\gamma$-phase lattices were formed by the phase transformation from $\alpha'$ to $\gamma$ since the nitrogen solute worked as a $\gamma$-phase stabilizer and as an alloying element in stainless steels [12]. In addition, the plastic straining was also induced to compensate for the strain

incompatibility between nitrogen saturated and unsaturated $\alpha'$-lattices [13]. The original coarse grains were refined by this large plastic distortion so that the nitrogen supersaturated layer consisted of an $\alpha'$-$\gamma$ two-phase, fine-grained structure [14,15].

In the present study, the fully martensitic stainless steel type AISI420 was plasma nitrided at 673 K for 3.6 ks to investigate the microstructure change from the surface across the nitriding front end to the depth of the matrix. The nitrided specimens were fine ground and polished down to 10 μm in depth for analysis in order to describe the transient zone from the nitrided layer to the matrix. EBSD (electron back-scattering diffraction) and SEM–EDX (electron dispersive X-ray diffraction) were first utilized to describe the microstructure change and the nitrogen content depth profile of the nitrided layer. In particular, TEM was also employed to describe the two-phase microstructure since the original coarse-grained microstructure was refined into an $\alpha'$- $\gamma$ two-phase layer. Across the nitriding front end, the original $\alpha'$-grains were aligned along the plastic slip lines with $\alpha'$ to $\gamma$-phase transformation in local. Most of the nitrogen supersaturated $\alpha'$-grains had a nanolaminated two-phase structure, where $\gamma$-phase nanolaminates were aligned with the width of 50 nm in the $\alpha'$-grain.

## 2. Experimental Procedure

An experimental setup for high density plasma nitriding at 673 K is explained with the standard processing conditions. How to prepare the samples for EBSD- and TEM-analyses is also stated with details on these analytical methods.

### 2.1. High Density RF–DC Plasma Nitriding System

A hollow cathode device was equipped to the radio-frequency–direct-current (RF–DC) nitriding system in order to intensify the nitrogen ion density. Figure 1 illustrates the experimental setup for the high density plasma nitriding process. A nitrogen–hydrogen mixture gas with the specified flow rate ratio was introduced to the inlet of the hollow. In the following experiments, a chamber in Figure 1 was evacuated down to the base pressure of 0.1 Pa after placing the specimen into the hollow. Nitrogen gas was only introduced into this chamber up to 300 Pa for heating.

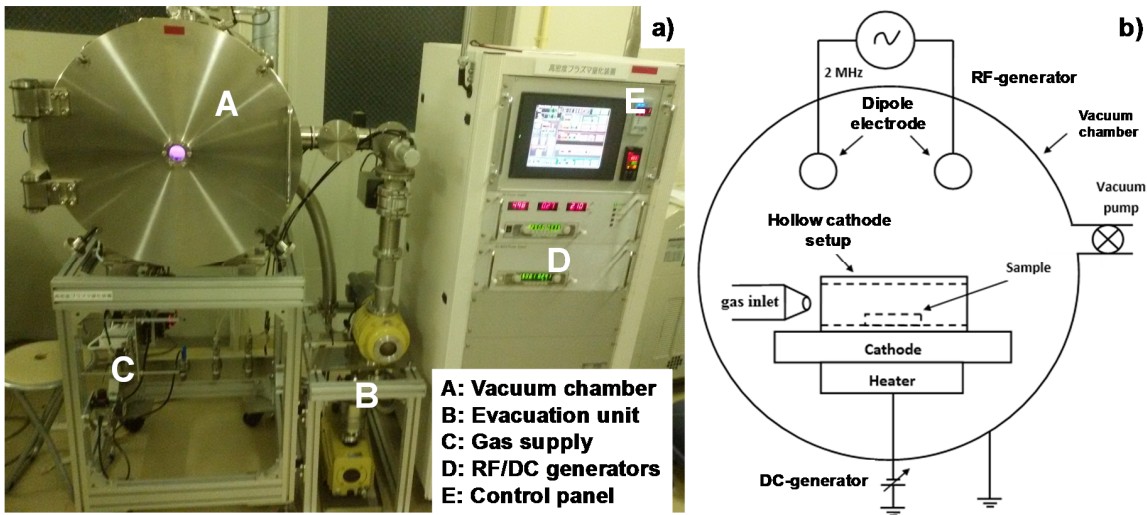

**Figure 1.** High density radio-frequency–direct-current (RF–DC) plasma nitriding system with use of the hollow cathode device; (**a**) the whole system; (**b**) illustration of experimental setup.

After the holding temperature became constant at 673 K, the pressure was controlled to also be constant by 35 Pa for presputtering. The presputtered specimen was further nitrided at 673 K for 3.6 ks by 70 Pa. During the presputtering and nitriding, both the pressure (P) and holding temperature (T) were automatically controlled to be constant with their deviations less than 0.1 Pa and 0.1 K, respectively. In addition, the input–output power balance was matched by the frequency adjustment

around 2 MHz. Furthermore, with use of the hollow cathode, as shown in Figure 1b, the plasma sheath with the high ion density in the order of $10^{18}$ ions m$^{-3}$ covered the whole specimen in this hollow during the presputtering and the nitriding processes afterward [16].

## 2.2. Preparation of Specimens and Samples

A martensitic stainless steel type AISI420 specimen with a size of 50 mm × 50 mm × 5 mm was prepared for this plasma nitriding. Its surface was mirror-polished to have a measured average roughness (Ra) of less than 0.1 μm. After plasma nitriding, the specimen was precisely ground and polished down to the depth of 10 μm to eliminate the surface nitrided layer. The polished surface had a lower Ra than 0.1 μm. This polished specimen was further cut into a sample with a size of 10 mm × 2 mm × 5 mm. SEM-EDX analyses were made on the top and side surfaces of the specimen, as shown in Figure 2a, with notes on the detection of nitrogen in the low energy spectrum.

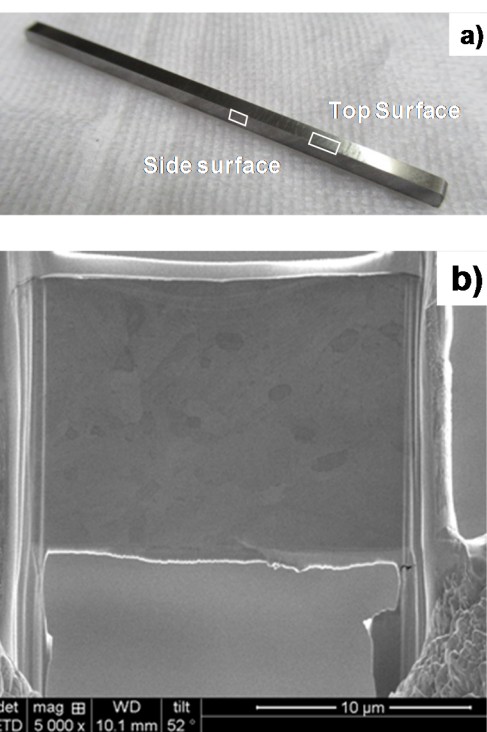

**Figure 2.** Preparation of specimen from the nitrided AISI420 substrate. (**a**) Ground and polished specimen for SEM-EDX (electron dispersive X-ray diffraction); (**b**) sliced specimen by FIB processing for TEM analysis.

## 2.3. EBSD Analysis

EBSD was utilized to provide analysis on the crystalline microstructure, the phase mapping, and the plastic straining from the surface to the depth in Figure 2a. The acceleration voltage was 15 kV, the area for analysis was 30 × 40 μm$^2$, and the spatial resolution was 0.05 μm. After EBSD analysis, the KAM (kernel average misorientation) distribution was measured to describe the plastic straining process during nitriding. Low KAM zones corresponded to the low plastic strain; higher KAM zones were plastically strained after [17]. Phase mapping by EBSD analysis also described the phase transformation during the nitriding. Furthermore, IPF (inverse pole figure) mapping provided the crystallographic microstructure with grain orientation.

## 2.4. TEM Analysis

TEM was used to make a precise microstructural analysis at the vicinity of the polished surface. A thin sample was prepared by slicing the original specimen in Figure 2a. SEM-FIB (Quanta 3D FEG;

FEI. Co. Ltd., Tokyo, Japan) was employed to make ion milling. In this milling, the acceleration voltage of Ga$^+$ ions, was constant by 30 kV for the normal milling process and reduced down to 2 kV for finishing. Figure 2b depicts a sliced sample for the present TEM analysis.

## 3. Experimental Results

The inner nitriding behavior and the microstructure change by nitriding in AISI420 substrate are described by SEM-EDX, EBSD, and TEM analyses.

### 3.1. Inner Nitriding Behavior in AISI420

The original AISI420 martensitic stainless steel substrates had no nitrogen content—even in trace levels—and an average grain size of 15 μm. Figure 3a shows the SEM image of the microstructure for the nitrided AISI420 substrate surface after being ground and polished by 10 μm from the initial surface. The original microstructure was refined to have a much smaller grain size. As depicted in Figure 3b, the nitrogen solute distributed uniformly on the ground and polished surface. The average nitrogen content analyzed by EDX reached to 5 mass% or 18 at%, equivalent to results in [14]. This high nitrogen concentration was characteristic to the nitrogen supersaturation.

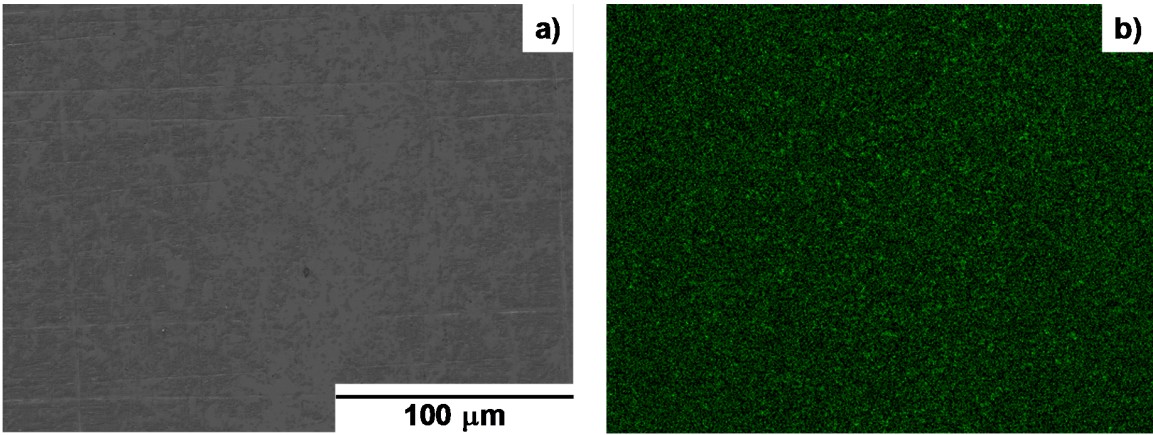

**Figure 3.** SEM image and nitrogen mapping on the ground and polished surface of nitrided AISI420 at 673 K for 3.6 ks; (**a**) SEM image; (**b**) nitrogen mapping by EDX.

Figure 4 depicts the microstructure and nitrogen mapping on the cross-section of the nitrided AISI420. The layer from the polished surface to the depth of 10 μm had a fine microstructure. As shown in Figure 4b, this layer had a much higher nitrogen content than that below the layer. Using the pointwise detection of nitrogen content in Figure 4b, the nitrogen content depth profile was analyzed by EDX (JOEL, Tokyo, Japan). As depicted in Figure 5, the nitrogen content gradually reduced from [N] = 5 mass% at the vicinity of the surface down to zero at the depth of 15 μm. This nitrogen content depth profile had influence on the microstructure across the nitriding front end.

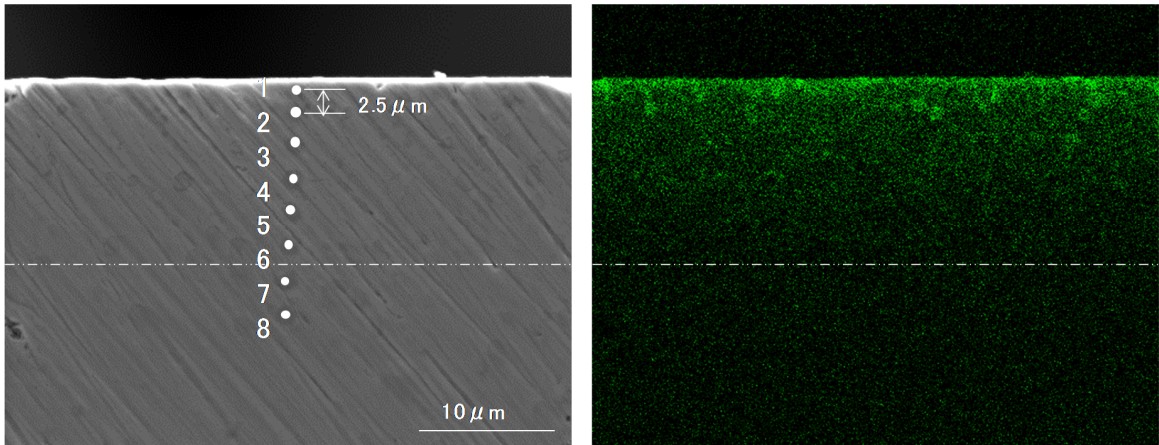

**Figure 4.** SEM image and nitrogen mapping on the cross-section of nitrided AISI420 at 673 K for 3.6 ks; (**a**) SEM image; (**b**) nitrogen mapping by EDX.

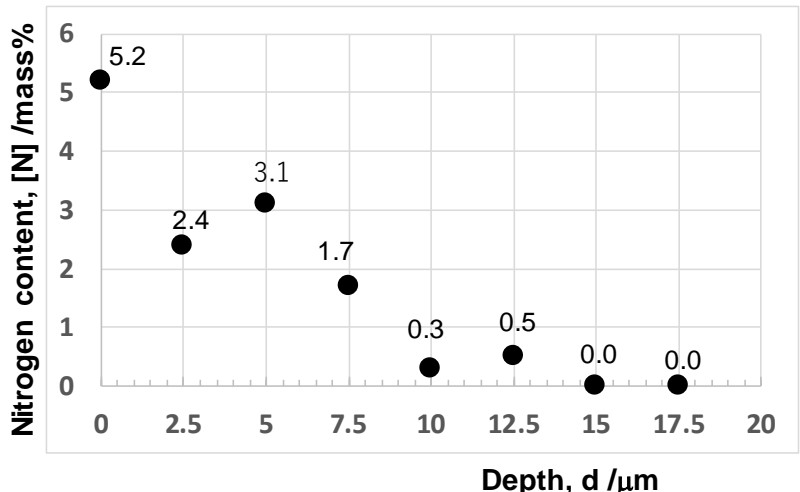

**Figure 5.** Nitrogen depth profile from the surface to the depth of nitrided AISI420 at 673 K for 3.6 ks.

### 3.2. EBSD Analysis on Nitrided AISI420

Figure 6a depicts the phase mapping on the cross-section of the nitrided AISI420 in correspondence to the SEM image and the nitrogen mapping in Figure 4. No $\gamma'$-Fe$_4$N or $\varepsilon$-Fe$_2$N precipitates were detected by XRD, even on the original surface. The original AISI420 had a fully martensitic phase with less retained austenites. The detected $\gamma$-phase by EBSD transformed from the original $\alpha'$-phase and distributed in the $\alpha'$-phase matrix. In the high nitrogen content layer with d < 10 μm in Figures 4 and 5, this $\gamma$-phase finely distributed with $\alpha'$-phase to form the two-phase microstructure. As in [14,15], $\alpha'$-lattices were elastically strained in expansion enough to make the phase transformation from $\alpha'$-phase to $\gamma$-phase. The surface layer down to d = 4 μm had a two-phase structure, while the two-phase zone grew only along the skewed direction into the depth for d > 4 μm. When d > 10 μm, the transformed $\gamma$-phases were detected in the $\alpha'$-phase matrix as larger zones without formation of the two-phase microstructure.

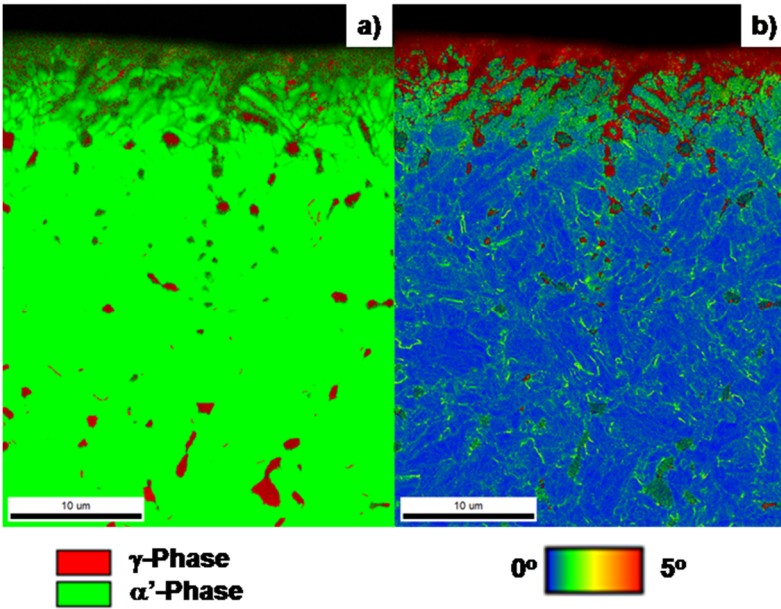

**Figure 6.** Phase mapping and kernel average misorientation (KAM) distribution on the cross-section of nitrided AISI420 at 673 K for 3.6 ks; (**a**) phase mapping; (**b**) KAM distribution.

KAM cross-sectional distribution is shown in Figure 6b. The high nitrogen content layer for d < 10 μm in Figure 5 had a higher misorientation angle than 5°. This revealed that the $\alpha' - \gamma$ phase transformation was accompanied with high plastic straining. In particular, a fine two-phase microstructure had a fine grain size by the plastic straining. That is, the original coarse $\alpha'$-grains were refined by plastic straining together with phase transformation to form a fine, two-phase structure. Below the two-phase layer for d > 4 μm in Figure 6a, a two-phase structure was only formed in the skewed directions along the high KAM lines. That is, phase transformation from $\alpha'$ to $\gamma$ localized at the highly strained slip lines.

In general, the plastic distortion consisted of the shear localization and spin rotation. The slip lines were formed from the surface to the depth by shear localization. The shape and size of grains were modified and reduced by the spin rotation. In the high nitrogen content for d < 4 μm, these slip lines were generated densely by shear localization, and the grain size was reduced. Whereas for d > 4 μm, the slip line density decreased with the depth, thus the phase transformation took place only at the cross-slipping points in Figure 6.

The inverse pole figure (IPF) distribution provided the information on the modification and reduction of the microstructure by the spin rotation in plastic distortion. Figure 7 depicts the IPF distributions from the surface to the depth in the normal direction (ND), the rolling direction (RD), and the tangential direction (TD), respectively. In the high nitrogen content layer with d < 4 μm, the original coarse $\alpha'$-grain with the average size of 15 μm was refined into finer grains with the average grain size of less than 0.5 μm. Even for d > 4 μm, most grains with large misorientation angles among neighboring grains had fine grain sizes ranging from 1 to 5 μm along the slip lines in Figure 6.

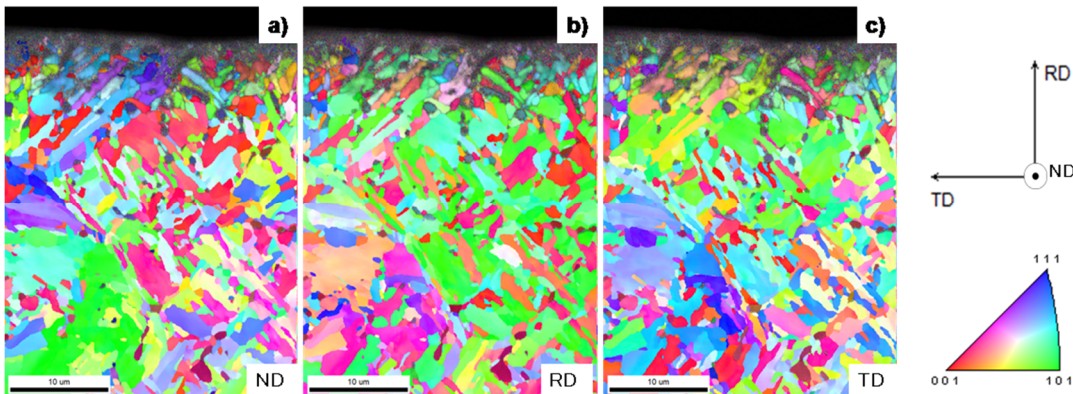

**Figure 7.** Inverse pole figure (IPF) distribution on the cross-section of nitrided AISI420 at 673 K for 3.6 ks in three directions; (**a**) IPF in the normal direction (ND); (**b**) IPF in the rolling direction (RD); (**c**) IPF in the tangential direction (TD).

In the depth for d > 4 μm, the grains in Figure 7 were geometrically shaped to be rectangular or rhombohedral. This revealed that the original α′-grains were sheared along the skewed slip lines and spin-rotated to align along the slip lines by plastic straining. In the high nitrogen content layer, the grain size was greatly reduced by this plastic localization to be less than the spatial resolution by EBSD. In the depth, the grains were reshaped and reduced to align along the slip line directions.

### 3.3. TEM Analysis on Nitrided AISI420

TEM analysis provided a means to describe the formation of the two-phase microstructure in the above by nitrogen supersaturation. Figure 8 shows the crystallographic structure at the vicinity of the polished surface across the nitriding front end down to d = 10 μm.

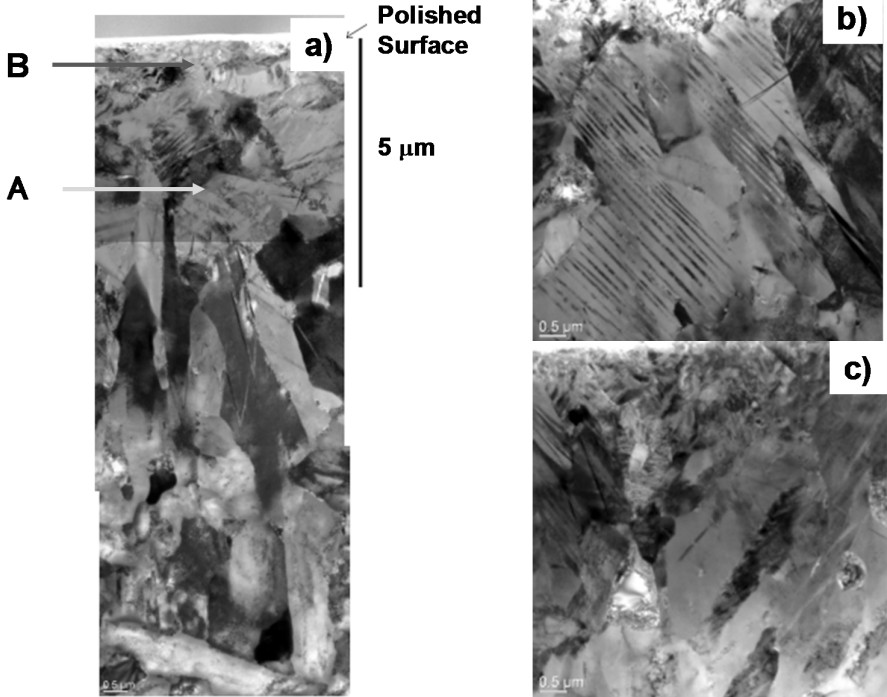

**Figure 8.** TEM image at the vicinity of the polished surface down to d = 10 μm; (**a**) overall crystallographic structure down to the depth of 10 μm; (**b**) microstructure at point A in the heterogeneously nitrided layer; (**c**) microstructure at point B in the transient to the homogeneously nitrided layer.

As shown in Figure 8a, the surface layer down to the depth of 2 μm was homogeneously nitrided to have fine grains in correspondence to the EBSD analysis in Figure 7. The transient zone from 2 μm to 5 μm consisted of the grains with sizes less than 5 μm, corresponding to the average grains size in the IFP mapping. To be noticed, these $\alpha'$-grains had a laminated microstructure in their insides. Across the nitriding front end for d > 5 μm, rectangular and rhombohedral grains were also seen, just as analyzed by EBSD in Figure 7.

Figure 8b depicts a typical microstructure at point A of d = 3 μm in Figure 8a. After EBSD analysis in Figures 6 and 7, this microstructure consisted of the nitrogen supersaturated $\alpha'$-grains as well as the two-phase grains. The laminated grain with the width of 50 nm corresponded to the two-phase grain. After precise analysis in [14], the volume fraction of $\alpha'$-phase was 70% in the nitrided layer. This formation of the laminated microstructure revealed that a whole $\alpha'$-grain did not transform fully to $\gamma$-phase, but $\gamma$-phase platelets with the width of 50 nm were formed into the $\alpha'$-phase zone in a periodic manner. That is, the two-phase nanostructure observed in Figures 6a and 7 was not a nano-mixture of $\alpha'$-grains and $\gamma$-grains but a nano-laminate of $\gamma$-zones into $\alpha'$-grain. To be noticed, these $\gamma$-phase platelets had a high aspect ratio of length to width.

Figure 8c shows the nanostructure at point B in the vicinity of the surface in Figure 8a. Most of the grains had finer sizes than those in Figure 8b. They also had nano-laminated microstructures with much narrower laminate widths than 50 nm in Figure 8b. Two-phase fine grains had the same nano-laminate structure, as seen in Figure 8b, irrespective of nitrogen content in the nitrided layer. The formation of the nano-laminated $\alpha'$-$\gamma$ two-phase microstructure is common to nitrogen supersaturated martensitic stainless steels.

## 4. Discussion

Few studies have reported on the phase transformation from the martensite to the austenite by the low temperature plasma nitriding of stainless steels. Let us discuss this unique phase transformation with reference to the literature on the materials science of iron and steels. In general, three phase transformation processes are reported in the heat treated and wrought steels, e.g., phase transformation from $\alpha$- to $\alpha'$-phase, from $\gamma$- to $\alpha'$-phase, and from $\alpha'$- to $\gamma$-phase, respectively [18]. In the huge literature of studies on the $\alpha$- to $\alpha'$-phase transformation, various morphologies of martensites have been reported by precise analyses [19]; e.g., a lath-martensite, where the specified zones in the $\alpha$-phase grain transform to martensite with the well-defined crystallographic orientations to $\alpha$-matrix. As seen in Figure 8, $\gamma$-nanolaminate in the two-phase grains looks like the lath-martensite in morphology. However, this $\gamma$-nanolaminate has a long aspect ratio with a laminate length of more than 5 μm and a width of 0.05 μm in Figure 8b against the crystallographic aspect ratio of lath-martensite with 33 to seven. In addition, no variants are seen in this $\gamma$-nanolaminate. This reveals that phase transformation into $\gamma$-nanolaminates by nitrogen supersaturation is not simply driven by the slip deformation process to drive the lath-martensitic transformation without the mass diffusion.

Most austenitic stainless steels transform to martensite by intense local plastic straining. For example, the piercing of a small hole into the AISI304 stainless steel sheets caused a massive transformation from $\gamma$- to $\alpha'$-phase with high plastic straining around the hole [20]. The massively transformed $\alpha'$-zones in the phase mapping were surrounded by the highly strained zones in the KAM distribution. In the reverse phase transformation of martensitic to authentic phases, by releasing the induced strains via the heat treatment, the $\gamma$-grains were also massively transformed from the $\alpha'$-grains. Hence, the present transformation into $\gamma$-nanolaminates was not driven by plastic straining or by releasing the strains in $\alpha'$-grains.

Let us consider the formation mechanism of $\gamma$-nanolaminate in the $\alpha'$-grain by nitrogen supersaturation. Figures 6 and 7 prove that phase transformation was accompanied by the plastic straining in the laminated $\alpha'$-grains. A small part of $\alpha'$-grain at the vicinity of its boundary was nitrogen supersaturated by local nitrogen diffusion from grain boundary. Then, this part expanded inside of the $\alpha'$-grain. The elastic strain energy by this expansion induced the $\alpha'$-$\gamma$ phase transformation.

The neighboring $\alpha'$-lattices to this $\gamma$-zone were plastically strained to compensate for the misfit distortion between the two zones. Since the nitrogen atoms diffused along the highly misfit-angled boundaries [21], more nitrogen solute was supplied through the existing zone boundaries to the front of $\gamma$-zone. Under further advancement of nitrogen supersaturation into the inside of $\alpha'$-grain, the elastic and plastic straining drove the continuous growth of $\gamma$-zones with the new boundary formations of $\alpha'$-$\gamma$ and $\gamma$-$\alpha'$ zones. This reaction resulted in the formation of $\gamma$-nanolaminates with high crystallographic misorientation angles into $\alpha'$-grain, as shown in Figure 8. In particular, this $\gamma$-nanolaminate formed in the $\alpha'$-grain together with the $\alpha'$-grain refinement by the nitriding in the high nitrogen content layer. As observed in Figure 8c, fine $\gamma$-nanolaminated $\alpha'$-grains formed at the vicinity of surface.

The micro-Vickers hardness testing was employed to evaluate the local hardness of the nitrided layer. The average hardness at d = 5 μm from the polished surface reached 1400 HV. The surface hardness decreased down to 1000 HV at 7 μm and finally to the matrix hardness at d = 10 μm. This revealed that the nano-laminated $\alpha'$-$\gamma$ structure had high strength and enough hardness to improve the wear resistance as a die and mold substrate material. The mechanical properties and performance were controlled by the $\gamma$-nanolamination into $\alpha'$-grain, as well as the two-phase grain size refinement.

## 5. Conclusions

In the low temperature plasma nitriding of martensitic stainless steel type AISI420, the nitrogen supersaturation advanced from its surface to its depth with the nitrogen diffusion process. The inner nitriding process was driven in two modes, i.e., the homogeneous nitriding followed the heterogeneous one. The homogeneous nitriding was characterized by the formation of a fine-grained, two-phase microstructure. The transformation from $\alpha'$- to $\gamma$-phase, the plastic straining, and the grain size refinement took place concurrently with nitrogen supersaturation under high nitrogen solute concentration. The homogenous formation of $\gamma$-nanolaminated two-phase fine-grains was driven by the nitrogen diffusion through the in situ-formed zone boundaries via the concurrent process of elastic and plastic straining into $\alpha'$-grains.

Toward the nitriding front end, the nitrogen supersaturation localized under lower nitrogen content. In this heterogeneously nitrided layer, the $\alpha'$ to $\gamma$-phase transformation localized along the slip lines by highly plastic straining. The original coarse $\alpha'$-grains were not refined much but reshaped into rectangular and rhombohedral grains by cross-slipping in plastic straining. The nitrogen diffusion was also only decelerated through these grain boundaries. Although the nitrogen supersaturation localized itself, the heterogeneous nitriding advanced under the similar relation between elastic and plastic straining to the homogeneous nitriding.

Nanolamination of $\gamma$-phase zones into $\alpha'$-grains was common to the phase transformation by homogeneous and heterogeneous nitriding, where the formation of $\gamma$-phase zones was accompanied by plastic straining under the nitrogen supersaturated state.

**Author Contributions:** T.A. developed the research plan together with S.-I.Y., made nitriding experiments and wrote this research article with T.Y. for fine EBSD analysis and K.M. for precise TEM analysis.

**Funding:** This study was financially supported in part by the METI (Ministry of Economy, Trades and Industries)-program on the supporting industries, 2018.

**Acknowledgments:** The authors would like to express their gratitude to H. Morita (Surface Design Engineering Laboratory, llc) for his help in experiments.

**Conflicts of Interest:** The authors declare no conflict of interest.

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
