# Peer review of "Microstructure of Plasma Nitrided AISI420 Martensitic Stainless Steel at 673 K"

_crystals, doi:10.3390/cryst9020060_

Round 1

Reviewer 1 Report

Dear Authors,

I have read Your manuscript with great attention and interest. In my opinion, the paper fills the gap existing in described subject and gives very important information for scientists and engineers. The submission falls within the scope of the journal and is sufficiently original and comprehensive, but I have a remarks, so I recommend it to publish after MINOR REVISION.

Remarks:

Title: change: “steels” to “steel”. One steel grade was investigated in the article.

The purpose of the work is not clearly defined.

Figure 1. write “Electrode”; “Generator”; “Chamber” and “Cathode” with a lowercase letters and “Gas” with a capital letter.

Why the nitriding time was set exactly at 3.5 ks? Was this due to any recommendations or from the earlier experiences of the authors?

Lines 82 and 86: What does the t symbol mean? Remove symbol 3 from unit of length.

Lines 83 and 85: Was the Ra value measured?

Figure 5: Add N content values over points and add values 2,5; 7,5; 12,5 and 17,5 in Depth axis.

Figure 7 is inserted too early. It should be inserted after sentence: “…from 1 μm to 5 μm.”

Line 227: ND, RD and TD abbreviations are not explained in the text but only in the caption of Figure 7.

Lines 268-270 and 309-311: Some journals do not recommend formulating such forward-looking conclusions

Line 273: Add literature sources after: “…plasma nitriding”

Line 312: Can the authors supplement the results with the microhardness distribution? Providing one value is valuable and interesting, but it does not create an opportunity to compare with the previously given research results.

Figures 1, 3, 4, 5, 6, 7 and 8: Add dots after numbers.

References:

Some references are very old (1960s), most of the authors cited are from Japan, and almost all from Asia. Modern principles of scientific objectivity require (if it is possible and justified) to refer to articles from around the world. In order to maintain the continuity of the scientific discussion, I can also recommend citing papers from Crystals or other journals of the MDPI publisher.

Editorial remarks:

Line 34: correct typo in “nitriding”, remove dash from “super-saturated”

Figure 6 caption: add space before unit K

Line 318: change “stemles” to “stainless”

Author Response

Relpy to reviewer-1 was attached in the following rebuttal file.

Reviewer 2 Report

The present study characterizes the formation of nitrided surface layer produced by supersaturation process by low temperature plasma nitriding.

Unfortunately, the manuscript has many grammatical and word-choice errors that make the content hard to understand, moderate English changes required. In addition, the study lacks sufficient research and citation (mostly national sources are cited, 50% of sources are self-citing!), which strongly reduces scientific soundness and significance. Not all of methods are adequately described, which gives a concern about accuracy of method applications and interpretation. For example, an EDX analysis of interstitial elements with small atomic number requires special SEM settings and has big statistic error, which should be taken into consideration. Further, the interpretation of EBSD data such as phase indication also needs verification, given by pattern indication accuracy (Confidence Index or similar). How precisely is the phase indication? Was also epsilon-phase taken into account? TEM investigation doesn’t provide any information on phase identification or distribution, since only grain shapes can be clearly interpreted from images. Of great concern is also, that discussion and conclusions are based on free interpretation of results. For example, the statement in row 301/302: “Since the nitrogen diffusion rate along the zone boundaries is faster than that through a’-matrix…” strictly needs a citation source, because, unlike a carbon, nitrogen does not possesses strong affinity to grain or phase boundaries.

Overall, the interpretation of results, discussion and conclusions do not reach an appropriate scientific level and must be improved.

Author Response

Reply to reviewer was included to the follwing file.

Round 2

Reviewer 2 Report

I happy to see the improvement of manuscript. Authors have succeeded in revision and correction, which results in higher quality of presentation and improved scientific soundness.

Author Response

SInce no comments and notes are given, I would like to make minor revision with notes to English statements.